# Quality control in oocytes by p63 is based on a spring-loaded activation mechanism on the molecular and cellular level

Daniel Coutandin[1,2,3†], Christian Osterburg[1,2,3†], Ratnesh Kumar Srivastav[1,2,3], Manuela Sumyk[1,2,3], Sebastian Kehrloesser[1,2,3], Jakob Gebel[1,2,3], Marcel Tuppi[1,2,3], Jens Hannewald[4], Birgit Schäfer[1,2,3], Eidarus Salah[5], Sebastian Mathea[5], Uta Müller-Kuller[6], James Doutch[7], Manuel Grez[6], Stefan Knapp[5,8,9], Volker Dötsch[1,2,3*]

[1]Institute of Biophysical Chemistry, Goethe University, Frankfurt, Germany; [2]Center for Biomolecular Magnetic Resonance, Goethe University, Frankfurt, Germany; [3]Cluster of Excellence Macromolecular Complexes, Goethe University, Frankfurt, Germany; [4]MS-DTB-C Protein Purification, Merck KGaA, Darmstadt, Germany; [5]Nuffield Department of Medicine, Structural Genomics Consortium, University of Oxford, Oxford, United Kingdom; [6]Georg-Speyer Haus, Frankfurt, Germany; [7]ISIS Neutron and Muon Source, Rutherford Appleton Laboratory, Didcot, United Kingdom; [8]Institute for Pharmaceutical Chemistry, Goethe University, Frankfurt, Germany; [9]Buchmann Institute for Molecular Life Science, Goethe University, Frankfurt, Germany

*For correspondence: vdoetsch@ em.uni-frankfurt.de

†These authors contributed equally to this work

**Abstract** Mammalian oocytes are arrested in the dictyate stage of meiotic prophase I for long periods of time, during which the high concentration of the p53 family member TAp63α sensitizes them to DNA damage-induced apoptosis. TAp63α is kept in an inactive and exclusively dimeric state but undergoes rapid phosphorylation-induced tetramerization and concomitant activation upon detection of DNA damage. Here we show that the TAp63α dimer is a kinetically trapped state. Activation follows a spring-loaded mechanism not requiring further translation of other cellular factors in oocytes and is associated with unfolding of the inhibitory structure that blocks the tetramerization interface. Using a combination of biophysical methods as well as cell and ovary culture experiments we explain how TAp63α is kept inactive in the absence of DNA damage but causes rapid oocyte elimination in response to a few DNA double strand breaks thereby acting as the key quality control factor in maternal reproduction.

## Introduction

The p53 protein family with its three members p53, p63 and p73 plays very important roles in the surveillance of genetic and cellular stability (*Levine et al., 2011*). Probably the most ancient function of this family is the maintenance of genetic quality in germ cells since even short lived eukaryotic animals express a p63-like protein in their germ cells (*Ollmann et al., 2000*; *Derry et al., 2001*; *Brodsky et al., 2000*; *Suh et al., 2006*; *Ou et al., 2007*). In mammals, up to 10 diverse p63 isoforms exist with the longest one, TAp63α, being highly expressed in primary oocytes that are arrested in prophase of meiosis I. After homologous recombination, oocytes are kept in this dictyate arrest phase until they are recruited for ovulation, a period that can take decades in humans. Once oocytes reenter the cell cycle, expression of TAp63α is lost (*Suh et al., 2006*). Since p63 can initiate

**eLife digest** The irradiation and chemotherapy drugs that are used to destroy cancer cells also damage healthy cells. Germ cells – from which egg cells and sperm cells develop – are particularly vulnerable as they contain sensitive quality control mechanisms that kill any cell that contain damaged DNA. Consequently, after surviving cancer many patients are confronted with infertility.

A protein called p63, which is closely related to another protein that suppresses the formation of tumors, plays an essential role in detecting and responding to DNA damage. In immature egg cells (also known as oocytes), p63 mostly exists in an inactive form. The protein then switches to an active form when DNA damage is detected to trigger the process of cell self-destruction.

Now, Coutandin, Osterburg et al. have performed a range of biochemical, biophysical and cell culture experiments to study how p63 is kept in its inactive form in the oocytes of mice. The experiments showed that in the inactive form, the two ends of the protein form a sheet that closes a key site on the protein and prevents it from changing into its active form. However, this closed form can be thought of as being like a spring-loaded trap – it doesn't take much energy to spring the trap and open the protein into its active form. Once this change has occurred, it is irreversible.

Coutandin, Osterburg et al. also found that the oocytes of mice already contain all the proteins necessary to activate p63. This means that once the switch to the active form is triggered there is no delay waiting for other proteins to be made, which makes oocytes extremely sensitive to DNA damage. Further work is now needed to investigate the exact molecular mechanisms behind the activation of p63.

apoptosis the high expression level of TAp63α in oocytes requires that its activity is tightly regulated. Recently we could show that TAp63α assembles into a closed and only dimeric conformation in which the protein is inactive (*Deutsch et al., 2011*). Detection of DNA damage leads to activation of p63 triggered by phosphorylation (*Suh et al., 2006*; *Bolcun-Filas et al., 2014*) that results in the formation of open tetramers with a twentyfold higher DNA binding affinity and the induction of apoptosis.

This p63-based quality control is unique to oocytes, making them very sensitive to DNA damage. Irradiation with 0.45 Gy is sufficient to eliminate all p63-expressing oocytes in mice while all surrounding cells of the ovaries survive. To understand the mechanism of inhibition and activation we have started to characterize the structural requirements for the formation of the closed and dimeric state of TAp63α. In previous experiments we have shown that the very C-terminus contains a transactivation inhibitory domain (TID) that is of central importance for creating the closed dimeric state (*Serber et al., 2002*; *Straub et al., 2010*). We have suggested a model in which both the C-terminal TID and the N-terminal transactivation domain (TAD) interact with the central tetramerization domain (TD) thereby preventing the formation of tetramers. This central TD is a dimer of dimers suggesting that blocking the interface by which two dimers form a tetramer is the most likely mechanism of inhibition. In the past we have identified mutations in all three domains – TAD, TD and TID – that break the inhibitory mechanism, establishing that at least these three domains are involved in this process. In the absence of a high resolution structure we have now used systematic alanine scanning and charge swap mutagenesis in combination with SAXS (small angle X-ray scattering) experiments to build a model of the closed and dimeric complex. In addition, we show that the inhibited conformation is a kinetically trapped state and that the oocyte contains all factors necessary to activate p63 without requirement of further protein expression. Together our data show that activation of TAp63α follows a spring-loaded mechanism and explains why oocytes are far more sensitive to DNA damage than the surrounding follicular cells.

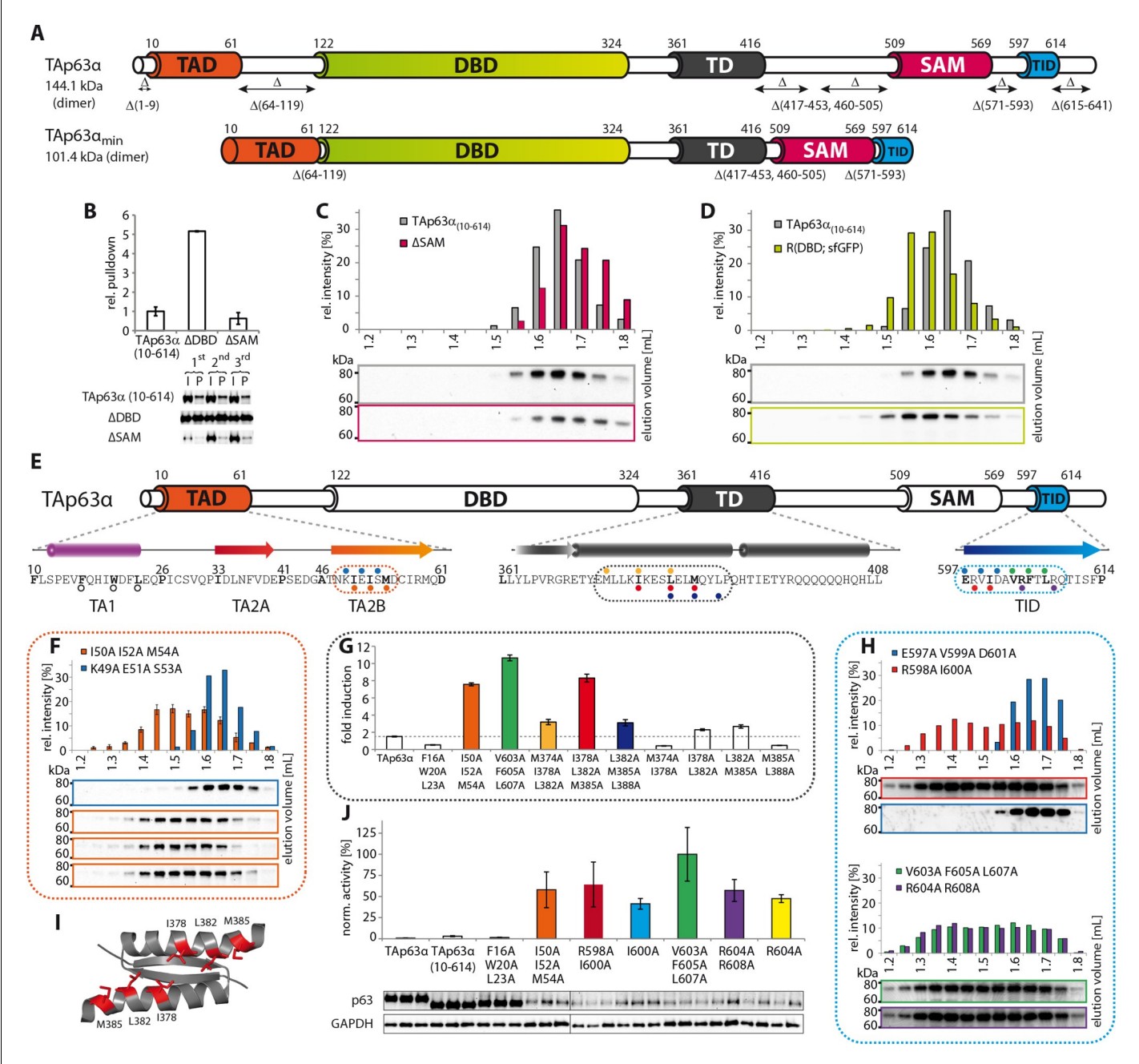

**Figure 1.** Mapping of structurally important regions within dimeric TAp63α. (**A**) Domain organization of TAp63α: transactivation domain (TAD), DNA binding domain (DBD), tetramerization domain (TD), sterile alpha motif (SAM) domain, transactivation inhibitory domain (TID). The minimal construct of TAp63α (TAp63α$_{min}$) lacks the first 9 and the last 27 amino acids as well as linker regions between TAD and DBD (64–119), TD and SAM (417–453; 460–505) and SAM and TID (571–593). Residues 454–459 were used as a linker between TD and SAM. (**B**) WB and corresponding bar diagram of pull-down experiments with constructs lacking either the DBD or the SAM domain using immobilized TID. Ratio of pull-down (P) and input (I) is shown relative to TAp63α$_{(10–614)}$ (set to 1). Pull-downs were performed in technical triplicates and error bars denote standard deviation. (**C,D,F,H**) TAp63α$_{(10–614)}$ constructs were expressed in rabbit reticulocyte lysate (RRL) and subjected to size exclusion chromatography (SEC). SEC profiles were obtained by WB (using an anti-myc antibody). (**C,D**) SEC profiles of TAp63α$_{(10–614)}$ ΔSAM (C; pink) and TAp63α$_{(10–614)}$ R(DBD; sfGFP) (D; green) compared with wild type (TAp63α$_{(10–614)}$, grey). R(DBD; sfGFP) indicates the replacement of the DBD by sfGFP. (**E**) Secondary structure prediction and mapping of structural motifs that stabilize the dimeric TAp63α. Cylinders and arrows represent α-helices and β-strands, respectively. Mutations (color-coded and indicated by filled circles) were introduced into TAp63α$_{(10–614)}$ on different faces of predicted secondary structure elements. The TAD is subdivided into TA1 (residues 10–26), TA2A (33–41) and TA2B (46–61). The TA1 forms an α-helix and the F16/W20/L23 motif constitutes the single interaction motif of the TA1. See *Figure 1—figure supplement 5* for a thorough mapping of the TA1. (**F**) The two faces of the β-stranded TA2B were mutated (residues i, i+2, i

*Figure 1 continued on next page*

*Figure 1 continued*

+4 to alanine). SEC profiles of I50A I52A M54A (orange) and K49A E51A S53A (blue). See *Figure 1—figure supplement 6* for a thorough mapping of the TA2. SEC of I50A I52A M54A was performed in technical triplicates and error bars denote standard deviation. (**G**) Transcriptional activities of TAp63α TD mutants on the p21 promoter in SAOS2 cells. Triple and double alanine mutations were introduced on the central hydrophobic interface of the TD. Bar diagrams show n-fold induction relative to the activity of the empty vector. Experiments were performed in biological triplicates and error bars denote standard deviation. (**H**) Mutations were introduced on the two faces of the TID β-strand. SEC profile of R598A I600A (red), E597A V599A D601A (blue), V603A F605 L607A (green) and R604A R608A (purple), Q609A I611A F613A (green) and R604A R608A (purple). See *Figure 1—figure supplement 7* for SEC profiles of other mutants. (**I**) Central hydrophobic interface of the dimeric TD, showing the important I378 L382 M385 motif. (**J**) Transactivation assay of TAp63α$_{(10-614)}$ mutants that appeared tetrameric in previous experiments (see **F**, **H** and *Figure 1—figure supplement 7*). Transcriptional activities on the p21 promoter in SAOS2 cells were normalized to the protein level (determined by WB and referenced on GAPDH level). Experiments were performed in biological triplicates and error bars denote standard deviation.

The following figure supplements are available for figure 1:

**Figure supplement 1.** Domains behave as pearls on a string in tetrameric p63.

**Figure supplement 2.** SEC-MALS proves the dimeric nature of TAp63α$_{min}$.

**Figure supplement 3.** Deletion of 322–342 does not disrupt the dimeric state.

**Figure supplement 4.** DBD is not essential to retain the dimeric state.

**Figure supplement 5.** The TA1 forms an α-helix.

**Figure supplement 6.** Mapping of structural motifs in the TA2.

**Figure supplement 7.** Mapping of structural motifs in the TID.

**Figure supplement 8.** Mapping of structural motifs in the TD by measurement of transcriptional activities.

**Figure supplement 9.** Validation of structural motifs by pull-down with GST-TID.

**Figure supplement 10.** Transcriptional activities of tetrameric TAp63γ mutants.

# Results

## Defining the minimal sequence required for formation of the closed dimeric conformation

TAp63α contains three folded domains, the DNA binding domain (DBD), the tetramerization domain (TD) and the SAM domain that are linked by unstructured regions. NMR experiments with a tetrameric construct containing all three folded domains showed that these domains behave independently as pearls on a string (*Figure 1—figure supplement 1*). All sequences outside of these folded domains are not structured in isolation but may be folded when interacting with other segments of the protein as part of the inhibitory mechanism. To identify the exact sequence elements required to form the closed state, we systematically deleted sequences in these linker regions. Deletion of sequences crucial for the formation of the closed state results in the formation of an open conformation. Previously we have shown that the open state can be detected by a conformation sensitive pull-down experiment: tetrameric mutants with an intact TAD can be pulled down with a GST-TID construct (569–616) (*Straub et al., 2010*). Thus, mutants that cannot be pulled down are assumed to adopt the closed dimeric state. After several rounds of deletion mutagenesis, a minimal dimeric construct was obtained. Size exclusion chromatography combined with multi angle light scattering (SEC-MALS) confirmed that this minimal construct (TAp63α$_{min}$) comprising deletions Δ(1–9; 64–119; 417–453; 460–505; 571–593; 615–641) is a stable dimer in solution (*Figure 1A* and *Figure 1—figure supplement 2B*). In addition, deletion of amino acids 322–342 between DBD and TD does not disrupt the dimeric state (*Figure 1—figure supplement 3*), but results in quite low expression levels in E. coli. For the experiments described below we have, therefore, used either TAp63α$_{min}$, wild type

TAp63α or a slightly shortened version TAp63α$_{(10–614)}$ lacking unstructured sequences in the N- and C-terminus (*Figure 1—figure supplement 2*).

## The SAM domain and the DBD are not essential to retain the dimeric state

In contrast to the TD, an involvement of the SAM domain and DBD in the formation of the closed dimeric state is not immediately obvious. To investigate whether these domains participate in the stabilization of the closed conformation we deleted each domain separately in TAp63α$_{(10–614)}$ and performed pull-down experiments with GST-TID. Interestingly, deletion of the SAM domain did not show any significant pull-down and size exclusion chromatography confirmed the formation of closed dimers (*Figure 1B and C*). On the contrary, deletion of the DBD resulted in a strong pull-down signal suggesting an open state (*Figure 1B*). Initially we expected the DBD to participate in essential domain-domain contacts that stabilize the closed conformation and therefore conducted an extensive mutagenesis screen of surface residues of the DBD (*Supplementary file 1*). However, none of the mutants formed tetramers making this hypothesis unlikely. Alternatively, the DBD may be important for geometric reasons, acting as a spacer between TAD and TD. To test this hypothesis, we replaced the DBD by superfolder GFP (sfGFP) which is very stable and of similar size as the DBD. SEC analysis of this chimeric protein expressed in rabbit reticulocyte lysate (RRL) suggested that it adopts a closed dimeric conformation (*Figure 1D*). Moreover, mutations F16A W20A L23A within the TAD and F605A T606A L608A within the TID resulted in the formation of a tetrameric state similar to experiments with wild type TAp63α (*Straub et al., 2010*) (*Figure 1—figure supplement 4C and E*). Similarly, replacement of the DBD by MBP enables the formation of a closed dimeric state (*Figure 1—figure supplement 4F*). These results suggest that the DBD does not participate in essential domain-domain interactions necessary to form the dimeric state and that the closed dimeric state of TAp63α is formed by interaction of the N-terminal TAD, the central TD and the C-terminal TID. Nonetheless, constructs that only contain these three domains did not form dimers but aggregated, suggesting that the DBD or a domain of similar size is necessary for structural reasons or for the folding process.

## Mapping of the TAD-TD-TID interaction

To build a first model of the closed state we used secondary structure prediction programs to identify potential secondary structure elements within the TAD and TID and alanine scanning in combination with SEC analysis to experimentally verify these predictions. The theoretical analysis predicted the existence of an α-helix in the TA1 region, two β-strands in the TA2A and TA2B regions of the TAD and a β-strand in the TID (*Figure 1E*). Alanine scanning of the TA1 confirmed that only mutations of F16, W20 and L23 that have previously been identified as crucial for binding of the TA1 to the TD (*Deutsch et al., 2011*), disrupted the closed conformation while mutations on the three remaining faces of the hypothetical helix had no effect (*Figure 1—figure supplement 5*).

To test the existence of the various β-sheets we mutated all amino acids on one side of each predicted β-strand to alanine (i, i+2, i+4). While mutations on both faces of the presumed first beta-strand (TA2A) did not affect the oligomeric state (*Figure 1—figure supplement 6B*), the mutations I50A I52A M54A located on one face of the predicted TA2B β-strand disrupted the dimeric state (*Figure 1F*). Alanine scanning of the TID showed that mutations on both sides of the presumed β-strand disrupt the dimeric state (*Figure 1H* and *Figure 1—figure supplement 7B*).

Stabilizing the dimeric state is most likely achieved by blocking the tetramerization interface of the TD and we also used alanine scanning of the TD to identify essential residues (*Figure 1—figure supplement 8*). Since mutations in the tetramerization interface that destabilize the dimeric state most likely also inhibit the formation of the tetramer, we did not use SEC analysis. Previously, we have shown that an open dimeric state is transcriptionally more active than the closed dimeric state (*Deutsch et al., 2011*). Mutating the hydrophobic amino acids I378, L382 and M385 alongside the second half of the α-helix of the TD led to high transcriptional activity as expected for an open conformation (*Figure 1G and I*, *Figure 1—figure supplement 8).*

We also used the measurement of the transcriptional activity as well as pull-down experiments with GST-TID to validate the results of our SEC analysis with the different alanine mutants (*Figure 1J* and *Figure 1—figure supplement 9*). As expected, all mutants that behaved like open and

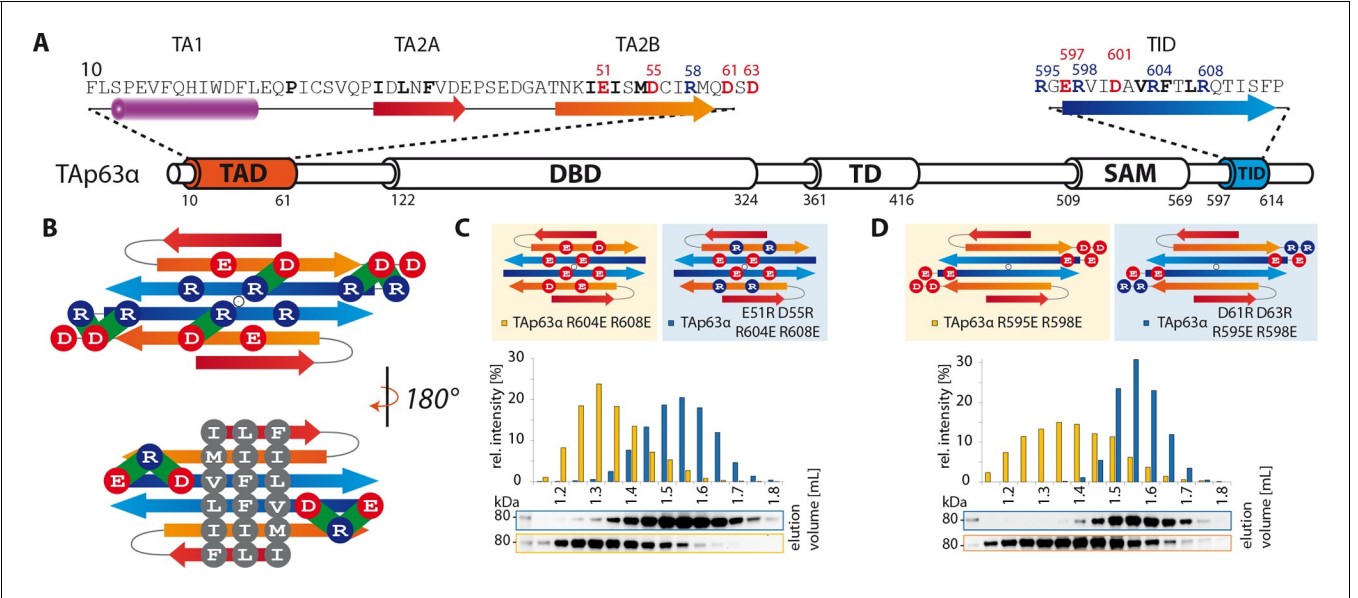

**Figure 2.** TA2B and TID form an anti-parallel β-sheet with a polar and a hydrophobic face. (**A**) Domain organization of TAp63α and secondary structure elements of TAD and TID. (**B**) Proposed interaction of TA2 and TID through β-sheet formation. This interaction is thought to be stabilized by hydrophobic amino acids clustered on one face of the β-sheet (bottom) and electrostatic interactions between charged amino acids on the other face (top). Extensive charge swap experiments (see *Figure 2—figure supplement 1*) revealed interactions between TA2B and TID. Interactions are depicted in green. (**C, D**) Introduction of negative charges in the TID and charge swaps between TID and TA2B show interaction via β-sheet formation. (**C**) SEC profiles of TAp63α R604E R608E (orange) and the charge swap mutant TAp63α E51R D55R R604E R608E (blue). (**D**) SEC profiles of TAp63α R595E R598E (orange) and the charge swap mutant TAp63α D61R D63R R595E R598E (blue).

The following figure supplement is available for figure 2:

**Figure supplement 1.** TA2B and TID form an anti-parallel β-sheet.

tetrameric conformations showed high transcriptional activity. The only exception was the F16A W20A L23A mutant since these mutations compromise the function of the TAD (*Figure 1—figure supplement 9*).

## TA2B and TID form a β-sheet

The experiments described above support the prediction that TA2B and TID form regular secondary structure elements, most likely β-strands. In the closed dimer, two TID and two TA2B sequences must be involved in the stabilization of the closed state. For symmetry reasons, the β-strands probably adopt an antiparallel orientation. Based on the results of the alanine scanning experiments we speculated that the two TID strands form the inner pair since mutations on both faces of the predicted β-sheet show strong effects. Further, we propose that the two TA2B strands form the two outer strands of a four stranded anti-parallel β-sheet which might be further extended by β-strands contributed by the TA2A segment. Such an arrangement would create one hydrophobic surface formed by I50/I52/M54 of TA2B and V603/F605/L607 of TID and a hydrophilic surface with residues E51/D55 of TA2B and R604/R608 of TID. The arrangement shown in *Figure 2B* brings charged amino acids on neighboring strands in close proximity, making it possible to test this hypothetical model by charge change and charge swap mutagenesis. Exchanging R604 and R608 in the TID to glutamic acids disrupted the dimeric state (*Figure 2C*). In our model these mutants created in combination with the negative charges on the TA2B strands a cluster of negatively charged amino acids that destabilized the dimer. Additional charge reversal of E51R and D55R in TA2B resulted in the formation of a stable dimer. Similarly, the R595E and R598E mutants are open tetramers and the additional charge reversal of D61R, D63R in TA2B rescued the dimer (*Figure 2D*). To refine our model and to identify the register of the proposed β-strands we used further pairwise charge swap mutations. The results of these experiments that all support our structural model are summarized in

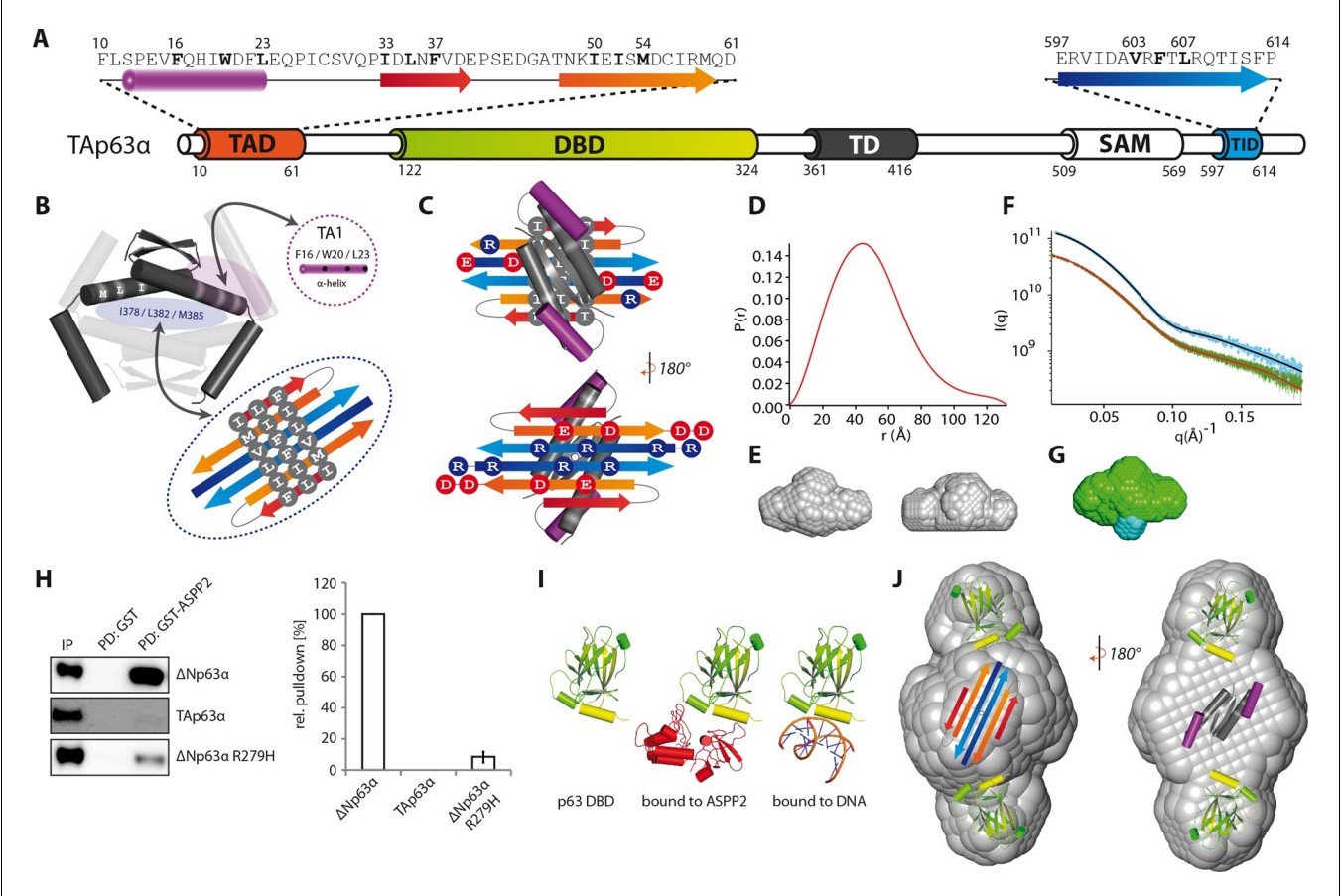

**Figure 3.** Model of the closed dimeric conformation of TAp63α . (A) Domain organization of TAp63α. All domains and structural elements are color coded. (B) The TD of p63 forms a dimer of dimers (colored in dark and light grey). Its two tetrameric interfaces (in light blue and rose) must be blocked in the inactive dimer to inhibit tetramerization. The TA1 was shown to bind to the upper interface (in rose) (*Deutsch et al., 2011*). The I378 L382 M385 motif in the central interface (in light blue) must be covered by hydrophobic amino acids. The hydrophobic interface of the proposed 6-stranded β-sheet is expected to cover this central tetrameric interface of the TD. (C) Model of the intramolecular interactions between TAD, TD and TID. The angles between structural elements are speculative. The TD was placed on top of the TA2/TID β-sheet so that the hydrophobic amino acids mask each other. The second helix of the TD is not modelled. (D) Pair distribution function P(r) from inline SEC-SAXS (small-angle X-ray scattering) data of TAp63α$_{min}$. Derived function transformed smoothly and appears to indicate globular central part with short extensional component. (E) Average ab-initio SAXS envelopes of TAp63α$_{min}$ without (left) and with (right) P2 symmetry, calculated using DAMMIF (*Franke and Svergun, 2009*). The similar shape suggests the presence of C2 symmetry in TAp63α$_{min}$. Envelopes were filtered and averaged using DAMFILT and were obtained from inline SEC-SAXS. (F) Simulated annealing multiphase model from simultaneous curve fits to wild type TAp63α$_{min}$ and λ-cro-TAp63α$_{min}$ (N-terminal fusion). Models constructed using MONSA allowing co-refinement of ab-initio models simultaneously. Blue segments give density differences derivative when refined against the native dataset. (G) Localization of the N-terminus. Multiphase fits to data sets, wild type TAp63α$_{min}$ in green and λ-cro-TAp63α$_{min}$ in blue. (H) WB and corresponding bar diagram of the pull-down experiments with ΔNp63α, TAp63α and ΔNp63α R279H from RRL using either immobilized GST or GST-ASPP2 fusion. WB signal for input (IP) and pull-down (PD) are shown. The pull-down efficiency of ΔNp63α was set to 100%. Pull-downs were performed in technical triplicates and error bars display the standard deviation. (I) Structure of the human p63 DBD alone, bound to DNA and a model of the p63 DBD bound to ASPP2 based on the co-crystal structure between the p53 DBD and ASPP2. (J) TAD, DBD, TD and TID are placed manually inside the P2 calculated average SAXS envelope. The DBDs are likely positioned at the outside of the molecule, leaving the center to be occupied by TAD, TD and TID. The SAM domain is not modelled.

*Figure 2—figure supplement 1.* Since the predicted β-sheet has one hydrophobic face and the interface used by the TD to form tetramers is also hydrophobic, we propose that the β-sheet covers the tetramerization interface of the TD, thus inhibiting the formation of tetramers (*Figure 3B and C*). In addition, the TA1 helix binds to the TD as well, further stabilizing the closed and compact conformation.

## Small angle X-ray scattering shows a dimeric structure of TAp63α with the DBDs at the outside

The mutational analysis described above predicted the formation of a compact structure with C2 symmetry. To verify this prediction, we performed SAXS measurements with TAp63α$_{min}$. To identify the localization of the N-termini we also collected SAXS data on a construct containing mutated λ-cro (Q27P, A29S, K32Q) at its N-terminus. Low resolution models derived from unbiased evaluation of the SAXS data showed indeed a C2 symmetry (*Figure 3E*) with the N-termini located in the center of the molecule (*Figure 3G*). Based on these results and the volumes of the individual domains we propose that the DBDs are positioned at the outside while the complex formed by the TAD, TD and TID builds the center of the molecule (*Figure 3J*). In this model the SAM domain is also located in the center where the molecule showed the largest volume.

To obtain additional information on the orientation of the DBD we performed binding studies with the Ankyrin Repeat and SH3 domain of the protein ASPP2. This protein is known to bind to the DNA binding interface of the DBD (*Figure 3I*). In pull-down experiments we were not able to detect interaction of TAp63α with ASPP2 while the open and tetrameric ΔNp63α isoform showed strong interaction (*Figure 3H*). This observation suggests that the DNA binding interface of the DBD is not freely accessible but points towards the core of the molecule.

## The dimeric conformation of TAp63α constitutes a kinetically trapped state

Activation of TAp63α entails breaking of the interactions described above to expose the tetramerization interface leading to the formation of active tetramers. In oocytes this transition is triggered by phosphorylation. In principle phosphorylation could provide a new interface contributing interactions that stabilize the tetrameric state, making it thermodynamically more stable while the dimeric state would be thermodynamically favored in the absence of phosphorylation. However, the observation that dephosphorylation of the open tetrameric state using λ-phosphatase does not result in converting TAp63α back to a dimer argues against this model (*Deutsch et al., 2011*). An alternative explanation would be that the tetrameric state is always the thermodynamically most stable one and the dimeric state is a kinetically trapped conformation. Phosphorylation would then function as a trigger to overcome a kinetic barrier and convert p63 into the thermodynamically preferred tetramer. Such spring-loaded mechanisms have been observed for example in the activation of influenza hemagglutinin (*Carr et al., 1997*; *Carr and Kim, 1993*). Characteristic for this type of activation mechanism is that perturbing the kinetically trapped conformation by moderate amounts of denaturants, changes in pH or an increase in temperature initiates the transition to the thermodynamically more stable conformation even without the natural trigger. Since the stability towards chemical denaturants of the three folded domains of TAp63α is quite high (*Klein et al., 2001*; *Sathyamurthy et al., 2011*) (*Figure 4—figure supplement 1*) we hypothesized that using low to moderate amounts of urea might disrupt the inhibitory structure, thus triggering the formation of the tetramer without affecting the folding of the DBD, the SAM or the TD. To investigate if activation of TAp63α follows a spring-loaded mechanism we equilibrated a SEC column with different concentrations of urea, incubated TAp63α$_{min}$ in buffer containing the same urea concentration and analyzed the percentage of dimer and tetramer. *Figure 4A* shows that a concentration of 1.75 M urea leads to an approximately 1:1 ratio of dimer and tetramer and at concentrations above 3 M no dimer was detected. Higher urea concentrations resulted in further shifts on the SEC column probably representing partially denatured conformations (*Figure 4—figure supplement 2*). To validate the data we performed SEC-MALS measurements at concentrations of 2 M and 2.5 M urea (*Figure 4F and G*). The first SEC peak had a mass of 197.9 ± 12.7 kDa (at 2.5 M urea) and the second peak a mass of 96.3 ± 6.4 kDa (at 2 M urea), consistent with the first one representing a tetrameric (202.8 kDa) and the second one a dimeric (101.4 kDa) conformation.

If the interpretation of the spring-loaded activation is correct, removal of urea would not allow the formation of a p63 dimer. To test this hypothesis, we separated the dimer and the tetramer fraction at a urea concentration of 1.75 M on the SEC column (*Figure 4C*) and dialyzed both fractions against buffer without urea. Re-analysis of these samples by SEC revealed that the dimeric fraction remained dimeric (*Figure 4D*) and the tetrameric fraction tetrameric with a tendency to aggregate

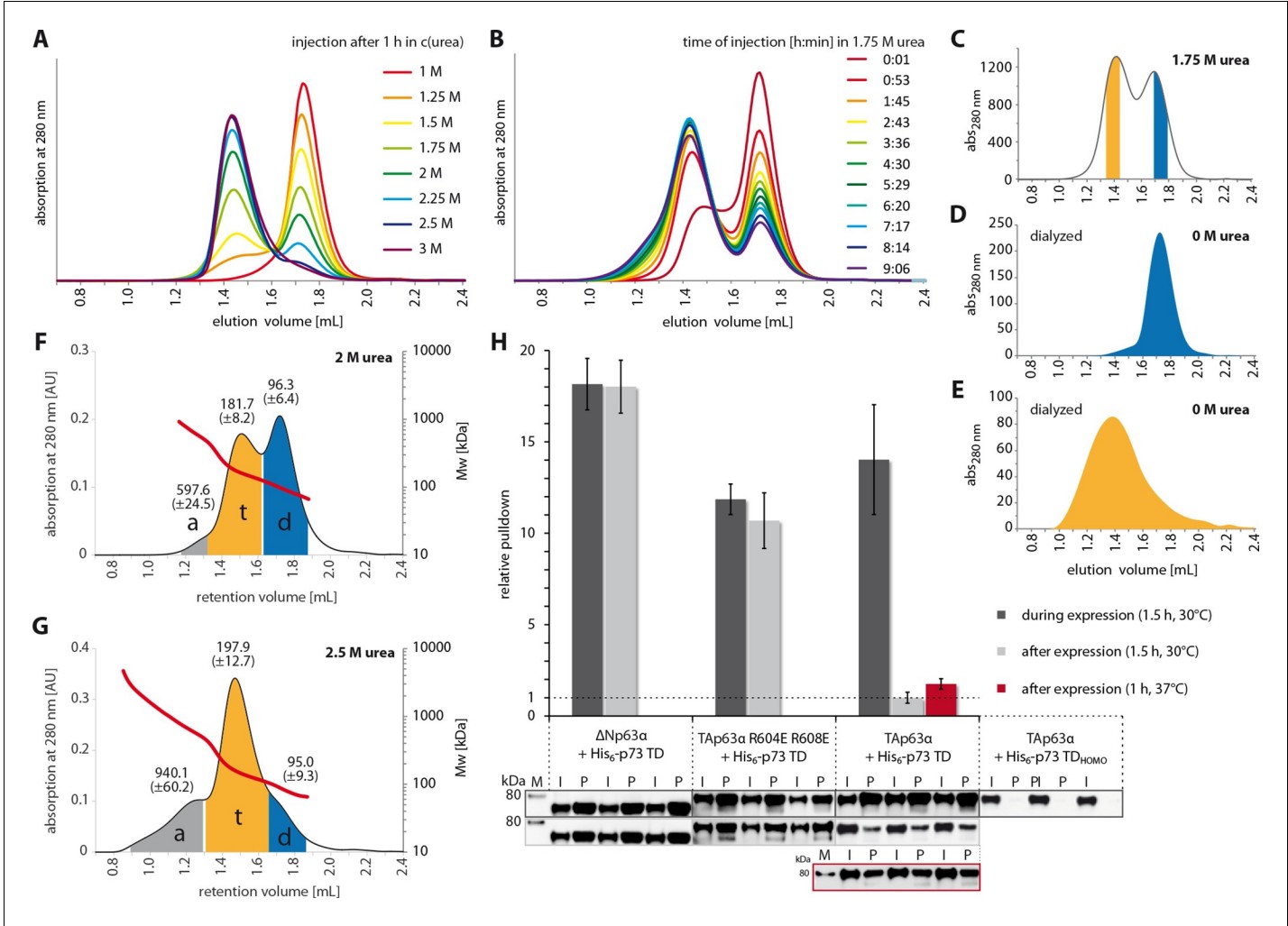

**Figure 4.** The closed dimeric conformation of TAp63α constitutes a kinetically trapped state. (A) TAp63α$_{min}$ samples were incubated for 1 hr at different urea concentrations and subjected to size exclusion chromatography (SEC) at corresponding urea concentrations. (B) TAp63α$_{min}$ samples were incubated in 1.75 M urea and injected into a Superose 6 3.2/300 column equilibrated with 1.75 M urea at different time points. (C) SEC profiles of TAp63α$_{min}$ injected after incubation for 50 min in 1.75 M urea. Fractions of tetrameric and dimeric protein are highlighted in orange and blue, respectively. (D,E) SEC profiles of reinjected tetrameric (E) and dimeric (D) fractions (originating from SEC shown in C) after dialysis to 0 M urea for 13 hr. (F,G) SEC-MALS of TAp63α$_{min}$ at different urea concentrations to proof the tetrameric nature of the early eluting peak in A. a, t and d denote aggregate, tetramer and dimer respectively. Colored areas where used to calculate the mean molecular weight and standard deviation. (F) SEC-MALS of TAp63α$_{min}$ in 2 M urea (preincubated in 2 M urea for 14 min at RT). (G) SEC-MALS of TAp63α$_{min}$ in 2.5 M urea (preincubated in 2.5 M urea for 25 min at RT). (H) WB and corresponding bar diagram of pull-down experiments with ΔNp63α, TAp63α R604E R608E and TAp63α incubated either during or after expression in RRL at 30°C for 1.5 hr with His$_6$-tagged p73 TD or a mutant that is not able to form hetero-tetramers (His$_6$-p73 TD$_{HOMO}$). Pull-down is achieved by hetero-tetramerization of His$_6$-tagged p73 TD with specified p63α constructs. Quotient of pull-down (P) and input (I) is shown relative to TAp63α incubated after expression with p73 TD (set to 1). Pulldowns were performed in technical triplicates and error bars denote standard deviation.

The following figure supplements are available for figure 4:

**Figure supplement 1.** Urea treatment of p63 structured domains.

**Figure supplement 2.** Urea unfolding experiments with TAp63α$_{min}$.

(*Figure 4E*). These experiments strongly suggest that the dimeric state of TAp63α is a kinetically trapped conformation that is activated by a spring-loaded mechanism.

## Formation of the TAp63α dimer can only be prevented co-translationally

A spring-loaded activation requires that the protein is trapped in a high energy state during protein synthesis. From p53 it is known that this protein forms dimers co-translationally (*Nicholls et al., 2002*), which in the case of TAp63α would enable the protein to fold into its closed conformation. To probe this hypothesis, we expressed TAp63α in RRL in the presence or absence of a high concentration (20 μM) of the isolated TD of p73. The rationale behind this experiment was that a high concentration of a domain that can interact with the TD of TAp63α during the translation would result in the formation of open tetramers. The TD of p73 was used since the isolated p63 and p73 TDs form hetero-tetramers that are thermodynamically even more stable than homo-tetramers (*Coutandin et al., 2009*). As a control we stopped the translation of TAp63α in RRL by adding cycloheximide (CHX) and then added the p73 TD to the same concentration as before and incubated for the same amount of time. Interaction between TAp63α and the p73 TD was monitored by pull-down experiments via the His-tag of the p73 TD. As shown in *Figure 4H*, expression in the presence of the p73 TD resulted in a strong pull-down while incubation post-translationally showed virtually no

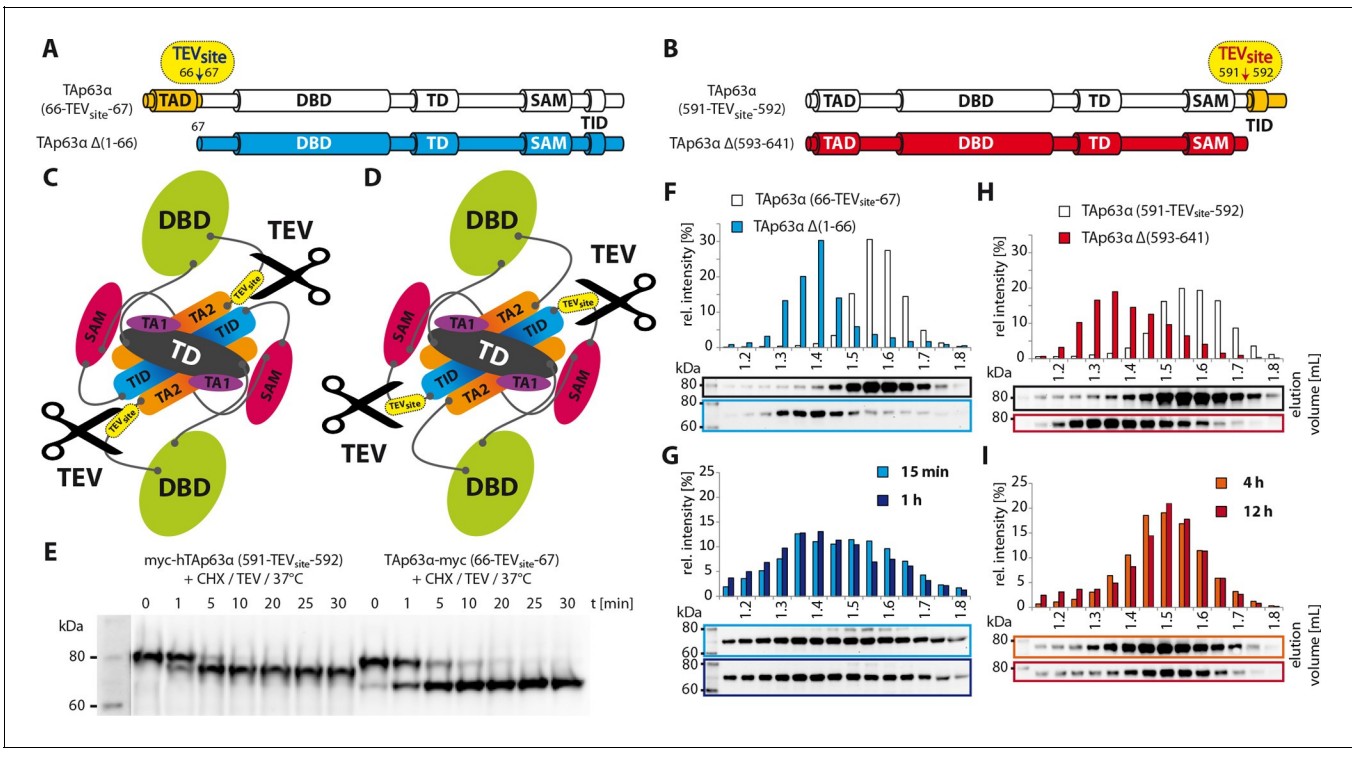

**Figure 5.** Unlike TID, secession of TAD induces the transformation of dimeric TAp63α to tetramers. (A) A cleavage site is introduced C-terminal to the TAD (between residues 66 and 67) allowing its secession by TEV protease cleavage. For comparison a TAp63α construct is created that lacks the TAD (TAp63α Δ(1–66)) and resembles the cleavage product. (B) A cleavage site is introduced N-terminal to the TID (between residues 591 and 592) allowing its secession by TEV protease cleavage. For comparison a TAp63α construct is created that lacks the TID (TAp63α Δ(593–641)) and resembles the cleavage product. (C) Schematic depiction of TAp63α (66-TEV_site-67) and secession of TAD by TEV protease cleavage. (D) Schematic depiction of TAp63α (591-TEV_site-592) and secession of TID by TEV protease cleavage. (E) Secession of TAD and TID from TAp63α derivatives using TEV protease. Cycloheximide (CHX) and TEV protease were added to the RRL expressed TAp63α derivative at 37°C and samples were taken after indicated time points and analyzed by western blotting. Both constructs are cleaved nearly completely within approximately 10 min. (F,G,H,I) TAp63α constructs were expressed in reticulocyte lysate (RRL), treated with CHX and optionally with TEV protease (G,I) at 37°C for denoted time, cooled to 4°C and subjected to SEC. SEC profiles were obtained by WB. (F) SEC profiles of TAp63α (66-TEV_site-67) and of TAp63α Δ(1–66). (G) SEC profiles of TAp63α (66-TEV_site-67) after treatment with CHX and TEV protease for either 15 min or 1 hr at 37°C. (H) SEC profiles of TAp63α (591-TEV_site-592) and of TAp63α Δ(593–641). (I) SEC profiles of TAp63α (591-TEV_site-592) after treatment with CHX and TEV protease for either 4 or 12 hr at 37°C.

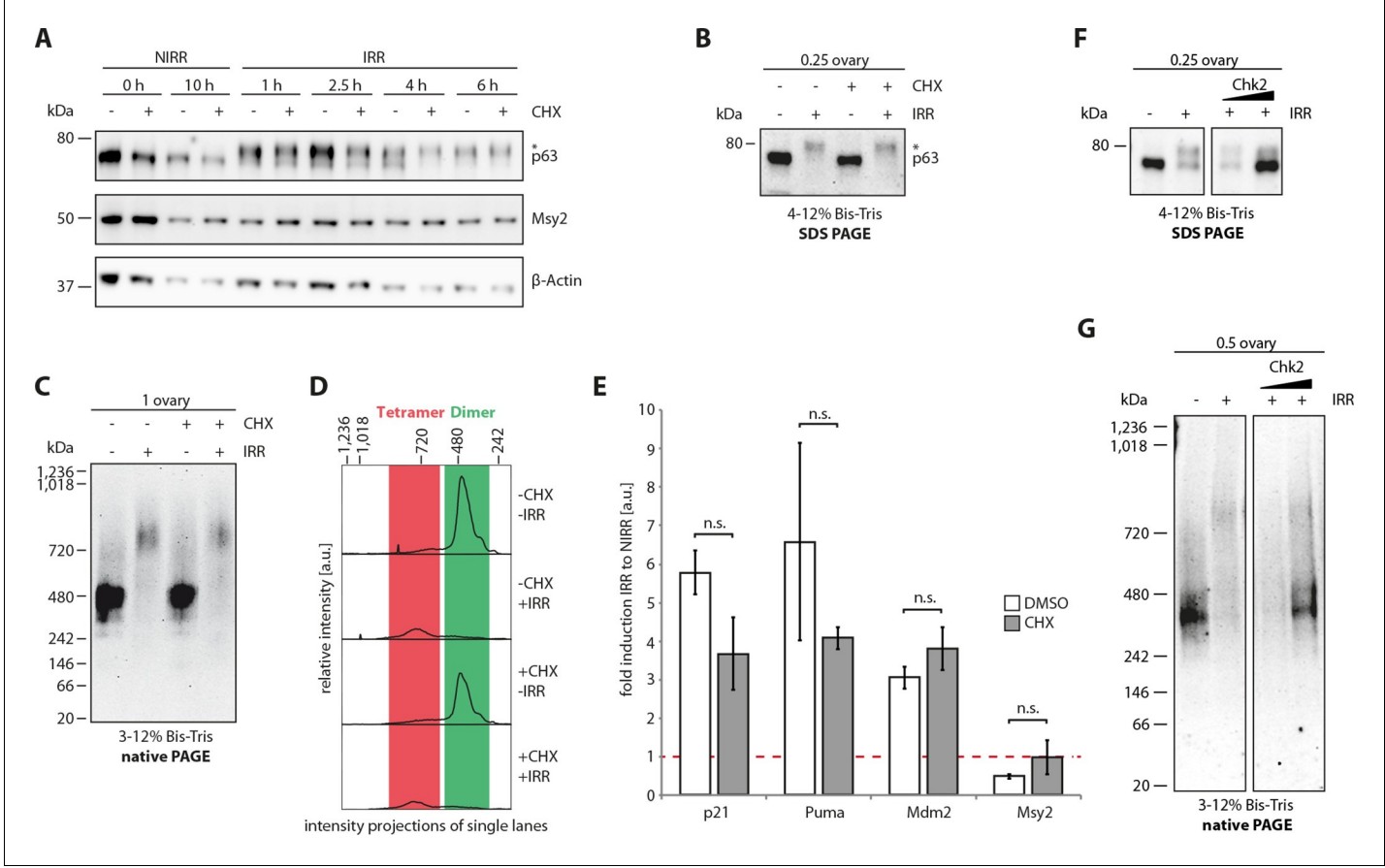

**Figure 6.** The cellular machinery for TAp63α activation in murine oocytes is always present and ready to act upon genotoxic insults. (**A**) WB of CHX treatment of nonirradiated (NIRR) and γ-irradiated (IRR) murine ovary samples. The signals of p63, the oocyte marker Msy2 and β-actin are displayed for each time point after NIRR/IRR. The asterisk marks phosphorylated p63. (**B**) WB of SDS-PAGE loaded with the ovary samples of the Native PAGE in (**C**). The asterisk marks phosphorylated p63. (**C**) WB of Native PAGE from (un-)treated and either NIRR or IRR murine ovaries. The p63 signal in the range from 20 kDa to 1,236 kDa is shown. (**D**) Intensity projection of the Native PAGE p63 signal from (**C**). The molecular weight range of the p63 dimer and tetramer is colored in green and red, respectively. (**E**) Quantitative Real-Time PCR of isolated murine oocytes. The bar diagram shows the fold induction of p21, Puma, Mdm2 and Msy2 mRNA after γ-irradiation. Error bars show the standard deviation of the biological duplicates. Brackets above the bars display the p-test results showing no significance (n.s.) between untreated and CHX treated oocytes for all targets. (**F**) Inhibition of Chk2 suppresses the DNA-damage induced phosphorylation of TAp63α in γ-irradiated ovaries. Chk2 inhibitor II at concentrations of 5 and 25 μM was added 2 hr before irradiation with 1.5 Gy. Ovaries were harvested 4 hr after irradiation and analyzed by SDS PAGE and Western Blot. Activated TAp63α gets degraded fast while preventing activation via inhibition of Chk2 preserves the original cellular concentration. (**G**) Native PAGE analysis of the same samples used as in (**F**). Inhibition of Chk2 prevents tetramerization and keeps TAp63α in a closed and dimeric state.

The following figure supplement is available for figure 6:

**Figure supplement 1.** p63 is responsible for inducing apoptosis in oocytes.

interaction with the p73 TD, even at elevated temperatures of 37°C. Replacing TAp63α in these experiments with open and tetrameric ΔNp63α or a tetrameric mutant TAp63α R604E R608E resulted in strong pull-downs both in the co-translational as well as in the post-translational setup. Performing the same experiments with a mutated TD that is not capable of forming hetero-tetramers showed no interaction. These results suggested that the kinetically trapped state of TAp63α is formed during or immediately after protein synthesis.

## The TAD defines the height of the kinetic barrier of trapped TAp63α

Oocytes survive the high concentration of TAp63α only when the inactivation mechanism is very effective. However, thermodynamics predicts that the closed conformation is always in equilibrium

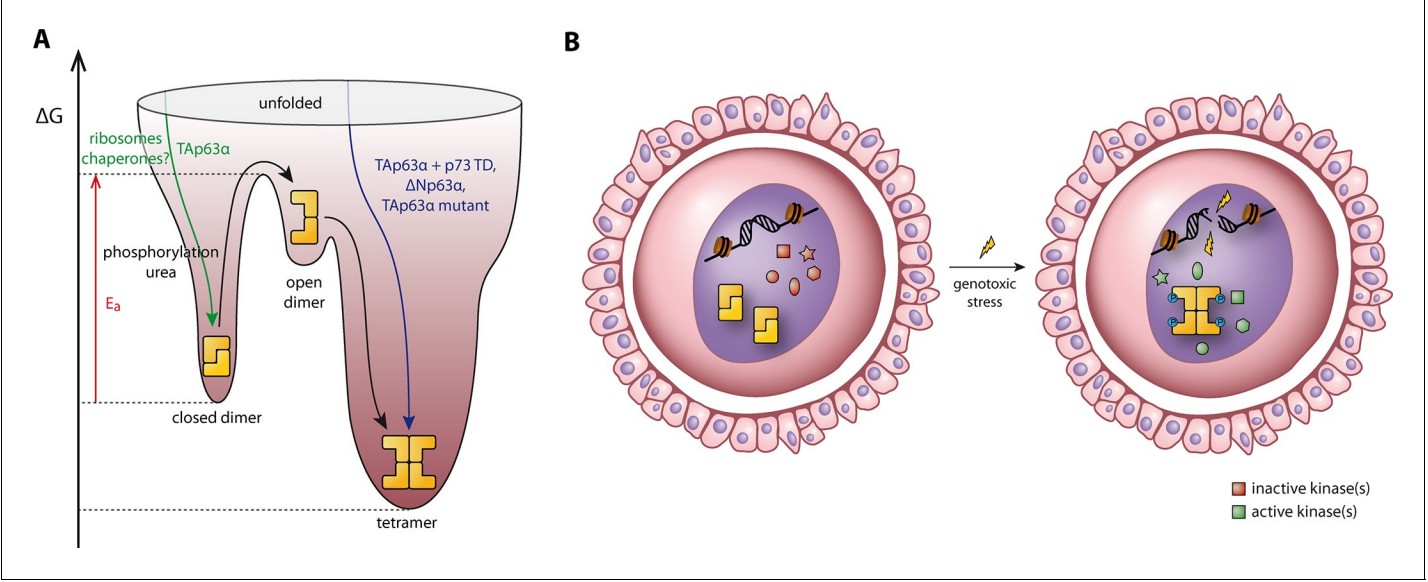

**Figure 7.** Spring-loaded activation mechanism of TAp63α on the molecular and cellular level. (**A**) Schematic energy landscape of TAp63α. The kinetically trapped closed dimer is opened by phosphorylation or artificially by moderate concentrations of urea (*Figure 4*). The resulting open dimer is less stable and forms tetramers with a dissociation constant of 12 ± 1 nM (*Brandt et al., 2009*). (**B**) Schematic representation of TAp63α activation. Oocytes express high levels of dimeric TAp63α and harbor normally inactive kinases ready to be activated and to phosphorylate TAp63α upon genotoxic stress leading to active tetramers and, consequently, cell death.

with more open conformations in which the inhibitory network of the TAD, TD and TID is at least partially broken. If during this partial unfolding no thermodynamically more stable tetramer is formed the dimer might be able to refold in its closed conformation. To obtain an estimation of the rate of unfolding of the TID and of the TAD we introduced TEV protease cleavage sites either C-terminal to the TAD or N-terminal to the TID. The rationale of this experiment was that after proteolytic cleavage the cleaved peptides (either the TAD or the TID) would diffuse away as soon as the p63 adopts an open conformation, therefore not allowing the protein to refold into its compact dimeric state and forcing it to form open tetramers. From this experiment the off rate of the corresponding domain can be estimated and thus the overall stability of the inhibitory lock mechanism. We incubated RRL expressed TAp63α with TEV protease for 15 min at 37°C which was sufficient to obtain close to 100% cleavage (*Figure 5*). The cleaved protein was then analyzed either immediately via SEC or further incubated for up to 12 hr at 37°C. Interestingly, cleavage near the TAD leads to the immediate formation of tetramers (*Figure 5G*). Unlike the TAD, the TID was bound with remarkable stability and cleaved p63 showed no tendency to assemble into tetramers even after long incubation times (*Figure 5I*). These results demonstrated that the N-terminus is the least stable part involved in keeping TAp63α dimeric and that its off rate determines the overall stability of the inhibited conformation. In addition, this interpretation further supports our model assuming that the TID forms the core of the central β-sheet.

## The oocyte contains the necessary machinery for the activation of p63 without protein expression

The experiments described above have demonstrated that TAp63α exists in a kinetically trapped state, poised to become activated upon the detection of DNA damage. Such a mechanism allows the cell to build an apoptotic switch with a sharp transition between survival and cell death. Indeed, measurements of the dose dependence of oocyte death have shown such a sharp transition with fewer than 10 double strand breaks per cell leading to oocyte death. To make such a system efficient the cell would need to be able to activate TAp63α fast which is best achieved when the activation machinery, i.e. the kinases required are already present and do not have to be expressed first. To investigate if oocytes have established such a pre-existing machinery, we harvested ovaries from

eight day old mice and γ-irradiated them with or without prior incubation with cycloheximide. Activation of TAp63α was followed by native geleletrophoresis. Addition of cycloheximide did neither prevent phosphorylation (*Figure 6A*) nor the formation of a tetrameric state (*Figure 6B–D*), suggesting that the kinases involved in detecting DNA damage and activating TAp63α are already present in resting oocytes. As a control to verify the effectiveness of the translation inhibitor cycloheximide we investigated the level of polyubiquitination (*Figure 6—figure supplement 1A*). Adding a proteasome inhibitor results in a strong accumulation of polyubiquitinated proteins that is suppressed by the addition of cycloheximide, as previously shown (*Mimnaugh et al., 2004*).

Induction of apoptosis requires the transcriptional activity of TAp63α and the translation of pro-apoptotic factors such as PUMA and NOXA (*Kerr et al., 2012*). To test whether the treatment with cycloheximide affects the transcriptional activity of TAp63α we used qPCR to detect mRNA levels of the three p63 targets p21, Puma and Mdm2 (*Figure 6E*). As a control we used the oocyte specific marker Msy2. The data showed that both with and without cycloheximide treatment significant induction of the target genes occurred while the level of Msy2 was unaffected. We could not detect the presence of p53 before or eight hours after irradiation by immunohistochemistry, suggesting that p53 is not involved in the apoptosis of oocytes (*Figure 6—figure supplement 1B*). This interpretation is also consistent with the observation that oocytes only from the TAp63 but not the p53 knock out mouse are protected from irradiation induced apoptosis (*Suh et al., 2006*). For p73 we could detect a weak, diffuse staining consistent with earlier reports of low levels of cytoplasmic p73 in oocytes (*Livera et al., 2008*). The very low level compared to p63 and the strong induction of target genes such as PUMA or p21 in the presence of the translational inhibitor cycloheximide, however, argue against a significant role of p73 for the irradiation induced cell death of oocytes.

Our results suggest that oocytes contain all kinases necessary to initiate tetramerization of TAp63α and all factors essential for p63's transcriptional function (*Figure 7B*). One of the kinases that has been identified in the activation process is Chk2 that phosphorylates TAp63α on Ser 582 (numbering according to the TA-isoform of p63) (*Bolcun-Filas et al., 2014*). To investigate if phosphorylation by Chk2 is required for tetramerization we treated mouse ovaries with increasing amounts of the Chk2 inhibitor II BML-277 and irradiated them with a dose of 1.5 Gy two hours after adding the inhibitor. At a concentration of 25 µM phosphorylation of TAp63α was almost completely suppressed and almost no tetramer was formed (*Figure 6F*). These data confirm the essential role of Chk2 in the activation process and demonstrate that phosphorylation by Chk2 is also a prerequisite for the formation of tetramers. Interestingly, these data also show that activation of TAp63α leads to a very significant drop of the intracellular concentration and inhibition of the activation by the Chk2 inhibitor to a preservation of the original level. This effect is due to fast proteasomal degradation of activated TAp63α and is consistent with other observations showing that the cellular concentration of active isoforms of p63 is low while inactive isoforms can accumulate to high concentrations (*Serber et al., 2002*). Interestingly, it has been shown that the N-terminal TAD is involved in this degradation process and that degradation is linked to DNA-binding competent and transactivating p63 isoforms (*Ying et al., 2005*). This observation is also consistent with our model in which the TAD is involved in the formation of the inhibitory lock structure that covers the tetramerizatoin interface and is therefore protected from ubiquitination. After the formation of the open and active state, however, the TAD is accessible, leading to fast degradation. This competition between activation and degradation probably constitutes an intracellular threshold that protect oocytes from apoptosis by low levels of activated TAp63α.

## Discussion

Oocytes are very special cells that have developed a unique quality control system. In humans the approximately seven million oocytes that are created during embryogenesis are diminished to one to two million at the time of birth (*Tilly, 2001*). A large drop in numbers is also seen for mouse oocytes. Of the original roughly 25,000 cells only 10,000 remain at the time of birth (*Di Giacomo et al., 2005*). During the late embryonic stage, the sensitivity of oocytes to DNA double strand breaks changes dramatically. While oocytes in the leptotene stage of prophase I (around E14) tolerate hundreds of Spo11 induced double strand breaks as part of the process of homologous recombination, postnatal oocytes are killed by fewer than 10 DNA double strand breaks per cell. This dramatic shift in sensitivity is correlated with the expression of TAp63α which starts to get expressed

in the diplotene stage beginning around E18.5 when chromosomes have been repaired after homologous recombination (*Livera et al., 2008*). Most likely, the p63 system developed as a safeguard to ensure that cells that still contain chromosome damage do not survive. The finding that the p63 expression level is kept high during the long dictyate arrest in mammals, however, shows that p63 is not only used as a short term quality control check point but also as a factor that guarantees the long term genetic stability of germ cells. In particular, this long term quality control function requires a tightly controlled activity of p63. A basal activity that is too high would lead to premature loss of the oocyte pool and ovary failure while a low activity bears the risk that oocytes acquire a high level of chromosomal defects. Our extensive mutagenesis study and biophysical characterization now provides a first model how interaction of N-terminal and C-terminal sequences blocks the tetramerization interface of the TD and therefore prevents tetramerization.

Our biochemical analysis has also revealed that neither the SAM domain nor the DBD are essential for formation of the closed state. However, unlike the SAM domain the DBD cannot be completely removed but must be replaced with a domain of similar size. At the same time, the ASPP2 binding assays in combination with the SAXS analysis predicts that the DBD has a defined orientation within the dimeric structure which makes the DNA binding interface inaccessible. This orientation, however, seems to be stabilized by interactions that are not essential for the formation of the core inhibitory structure consisting of the TAD, TD and TID. At the same time, this finding explains why both in cells as well as in in vitro fluorescence anisotropy measurements the DNA binding affinity of TAp63α is roughly 20-fold lower than the affinity of open and tetrameric isoforms or mutants (*Deutsch et al., 2011*; *Suh et al., 2006*).

Mutations in the SAM domain as well as in the TID cause the Ankyloblepharon-ectodermal defects-cleft lip/palate syndrome (AEC) syndrome (*McGrath et al., 2001*). Two mutations identified in human patients, R598L and D601V (*Rinne et al., 2009*), are located in a region of the TID that is responsible for stabilizing the dimeric state. According to our model both R598 and D601 are involved in charge-charge interactions with the TA2B β-strand and their mutation likely destabilizes the closed dimeric state which might cause in addition to the severe skin phenotype of the patients further ovary related problems. Mutations found in other domains of p63 such as the DBD that cause ectrodactyly–ectodermal dysplasia–cleft (EEC) syndrome (*Celli et al., 1999*; *Kouwenhoven et al., 2015*) might also affect the stability of the closed dimer in oocytes.

While effective inhibition is a prerequisite for a stable long term quality control with a minimal protein turnover rate, an effective activation mechanism is also of paramount importance. Our results show that the closed conformation of TAp63α is a metastable state and that activation follows a spring-loaded mechanism (*Figure 7A*). In oocytes, phosphorylation is used as the natural trigger to initiate the transition from the closed dimeric state to the thermodynamically more stable tetrameric state. Once the active tetramer is formed, the phosphate groups can be removed without affecting the oligomeric state of the protein (*Deutsch et al., 2011*). Spring loaded activation mechanisms are known from other proteins as well. One prominent example is the Influenza virus hemagglutinin A (HA). This membrane protein is trapped in a metastable native pre-fusion state in which the fusion peptide is buried inside the trimeric structure (*Carr and Kim, 1993*). Following endocytosis of the virus and a pH drop in the endosome, the protein changes its conformation resulting in the exposure of the fusion peptides that are subsequently inserted into the host membrane (*Lin et al., 2014*). While the drop in pH is the natural trigger, activation can also be initiated by high temperatures or urea (*Carr et al., 1997*). Another example is α-lytic protease, a secreted serine protease that is expressed with an N-terminal pro-region that catalyzes folding from a stable molten globule-like intermediate. Proteolytic degradation of the pro-region results in release of the native and active protease, which is thermodynamically less stable than the partially unfolded state but remains folded due to a large barrier to unfolding (*Sohl et al., 1998*; *Baker, 1998*).

The kinetically trapped state of dimeric TAp63α raises the question how and when this state is formed during protein synthesis. Interestingly, it was shown that p53 forms dimers co-translationally and tetramers post-translationally (*Nicholls et al., 2002*). Our expression experiments in the presence of high concentrations of the p73 TD in principle support a co-translational folding of TAp63α. However, our deletion mutagenesis also implicates that the last amino acid of TAp63α$_{min}$, P614, has to emerge from the ribosomal exit tunnel before the closed dimeric state can be formed. As a model we propose that open dimers form co-translationally via the TD that acts as the interaction platform

for the TAD and the TID to fold into the trapped conformation after completion of translation. The exact mechanism of folding and a potential role for chaperones remains to be investigated.

Not only the metastable state of TAp63α sensitizes oocytes for DNA damage induced cell death, the entire machinery that detects DNA damage and activates TAp63α is present in resting oocytes without the need for further protein expression. So far, ATM/ATR as upstream kinases and Chk2 as a direct phosphorylating kinase have been shown to be involved in this process (*Bolcun-Filas et al., 2014*; *Kim et al., 2011*). Other factors might contribute as well (*Gonfloni et al., 2009*) in stabilizing the tetrameric state and forming active transcriptional complexes on promotor sites. The special metabolic state that oocytes reside in during dictyate arrest requires them to express a limited number of genes, essential for keeping the cells stable. Proteins involved in the surveillance of DNA damage as well as transmitting the signal to the central integrator, p63, are part of this cellular repertoire. Quality control in oocytes by TAp63α is therefore based on a spring-loaded activation mechanism on the molecular and the cellular level.

## Materials and methods

### Expression and purification in E. coli

TAp63α was codon-optimized for expression in E. coli and ordered from Genscript (Piscataway, NJ, USA). Deletions were introduced using the QuikChange II Site-Directed Mutagenesis Kit (Agilent Technologies). TAp63α$_{min}$ comprising deletions $\Delta$(1–9; 64–119; 417–453; 460–505; 571–593; 615–641) was cloned into pNIC28-Bsa4 (SGC Oxford) by ligation independent cloning (*Gileadi et al., 2008*). The protein, bearing a N-terminal His$_6$-tag and a TEV (tobacco etch virus) protease cleavage site was expressed in BL-21(DE3)-R3-Rosetta (SGC Oxford) and initially purified using Ni-Sepharose Fast Flow and HiTrap Q HP (GE Healthcare) according to standard protocols. After His$_6$-tag removal using TEV protease the protein was further purified using a HiTrap Q HP and a HiLoad 16/600 Superdex 200 prep grade column. TAp63α$_{min}$ was stored concentrated (100 mg/mL) at -80°C.

### GST-ASPP2 expression and purification

ASPP2 (891–1128) was cloned into pGEX 6p2 (GE Healthcare) with an additional C-Terminal His$_6$-tag. The resulting GST-fusion of ASPP2 was expressed in BL-21(DE3)-R3-Rosetta (SGC Oxford) and purified by Ni-Sepharose Fast Flow and Gluthation-Sepharose Fast Flow (GE Healthcare) using standard protocols followed by size-exclusion chromatography with a HiLoad 16/600 Superdex 200 prep grade column.

### Multi-angle light scattering (MALS)

SEC-MALS experiments were performed at room temperature using a Superose 6 3.2/300 column (GE Healthcare) in phosphate buffer containing 0, 2 or 2.5 M urea on an Agilent 1200 Series HPLC system at a flow rate of 0.05 ml/min. Prior to injection the protein was incubated in phosphate buffer containing 2 M urea for 14 min or 2.5 M urea for 25 min. Elution of 10 μL of purified proteins of 6.4 mg/ml concentration was detected using Dawn Heleos II (11 angles were used) and an Optilab rEX Refractive Index Detector at a Laser wavelength of 658 nm (Wyatt Technology) to determine the weight average molar mass MW of peak locations. Data were processed using ASTRA software package 6.1.2.84 (Wyatt Technology).

### Native PAGE

For Native PAGE analysis of the oligomeric state of p63 two ovaries per indicated condition were harvested in 20 μl of ice-cold lysis buffer A (50 mM Tris pH 8.0, 100 mM NaCl, 1 mM DTT, 2 mM MgCl$_2$, supplemented with 1x cOmplete and PhosSTOP (Roche)). Lysis was performed by mechanical force using a pestle, pipetting and two cycles of freeze and thaw. After addition of 20 μl lysis buffer B (lysis buffer A containing 40 mM CHAPS) and 1 μl benzonase, samples were incubated for 1 hr on ice and subsequently centrifuged for 10 min at 4°C and 13.2 krpm to remove cell debris. 20 μl of supernatant were supplemented with 5 μl of 5x Native PAGE sample buffer (60% glycerol, 25 mM coomassie G250) for Native PAGE analysis. The remaining lysate was used for analysis of p63 level and phosphorylation-induced mobility shift via SDS-PAGE.

The separation of ovary lysate by Native PAGE followed by detection of p63 via subsequent Western Blot analysis was performed with the Native PAGE Novex 3–12% Bis-Tris protein gel system (Life Technology) according to the manufacturer's instructions. The cathode buffer was supplemented with 0.002% coomassie G250 and the separation was performed at 4°C for 60 min at 150 V and 90 min at 250 V.

## NMR spectroscopy

For NMR spectroscopy [u-$^{15}$N]-labeled human p63 DBD-TD-SAM, DBD, TD and SAM were measured at concentrations between 0.1–0.3 mM in a total volume of 350 µL in shigemi NMR tubes. Complete Protease Inhibitor (Roche) and 6% of a D$_2$O/DSS (3 mM DSS) solution was added. NMR-Experiments were performed on a Bruker Avance spectrometer equipped with $^1$H triple resonance, z-gradient cryogenic probes at a proton frequency of 900 MHz. All experiments were performed at 303 K. DSS (4, 4-dimethyl-4-silapentane-1-sulphonate) was used as an internal chemical shift reference. Spectra were processed with Bruker Topspin 2.1 and analyzed with UCSF SPARKY 3.114 (*Kneller and Kuntz, 1993*).

## Ovary culture

Animal care and handling were performed according to the guidelines set by the World Health Organization (Geneva, Switzerland). Eight-day-old (P8) female CD-1 mice were purchased from Charles River Laboratories. Ovaries were harvested, transferred in sterile flat-bottom 96-well plates with 100 µl MEM (+ L-Glu, Gibco) supplemented with 5% FBS, 0,4% BSA (w/v), Pen/Strep and 70 µM Br-cAMP and cultured in an incubator at 37°C with 5% CO$_2$.

Ovaries were treated overnight with either DMSO or CHX (50 µg/mL) prior following experiments. IRR ovaries were exposed to 1.5 Gy of γ-irradiation on a rotating turntable in a $^{137}$Cs irradiator, at a dose rate of 2.387 Gy/min. For inhibition of Chk2 in ovary culture the inhibitor BLM-277 (Merck Millipore, 220486) was used 2 hr prior γ-irradiation in indicated concentrations.

The following antibodies were used for detection of endogenous protein of ovary samples by Western Blotting: Msy2 (Santa Cruz, N-13), Ubiquitin (Santa Cruz, P4D1), p63 (Santa Cruz, H-129) and β-Actin (Santa Cruz, C4).

## Immunohistochemistry (IHC)

Dissected ovaries were cultured overnight and subsequently treated with γ-irradiation as indicated. Ovaries were fixed in formalin, embedded in paraffin and sectioned into 6 µm thickness (Morphisto GmbH, Frankfurt, Germany). For 3,3'-Diaminobenzidine (DAB) IHC staining sections were deparaffinised and rehydrated followed by 30 min antigen retrieval in boiling 0.1 M citrate buffer. Sections were blocked for 1 hr at room temperature in 5% donkey normal serum (Santa Cruz, sc-2044) in TBS and incubated with primary antibody raised either against the oocyte marker Msy (Santa Cruz, N-13, 1:200), p53 (Santa Cruz, DO-1, 1:100), p63 (Santa Cruz, H-129, 1:200) or p73 (Merck Millipore, ER-15, 1:100) in 1% BSA in TBS overnight. Sections were developed after incubation with biotin-conjugated secondary antibodies for 1 hr at room temperature in 1% BSA in TBS (Vector Labs) with the ABC DAB Peroxidase System (Vector Labs). Nuclei were stained for 5 min in Mayer's hematoxylin followed by dehydration and mounting of the stained sections.

## Protein expression in rabbit reticulocyte lysate (RRL)

N-terminally myc-tagged human TAp63α, TAp63α$_{(10-614)}$, TAp63γ, ΔNp63α and all mutants that base on these constructs were expressed from pcDNA3.1 vector in RRL as described (*Straub et al., 2010*). Proteins were used for SEC analysis and pull-down experiments.

## Pull-down experiments

GST pull-down experiments were performed with RRL expressed proteins and immobilized GST-TID (aa 569–616) as described previously (*Straub et al., 2010*).

## Pull-down experiments with His$_6$-tagged p73-TD

For His$_6$ pull-down experiments, ΔNp63α, TAp63α and TAp63α R604E R608E were expressed in presence or absence of His$_6$-tagged p73 TD (20 µM) in 50 µL RRL for 90 min at 30°C. In the latter

case, cycloheximide (50 µg/mL final) and His$_6$-tagged p73 TD (20 µM final) were added after expression and incubated for another 90 min at 30°C. Afterwards 5 µL samples were removed as input controls (I). For each pull-down 50 µL Ni-IDA beads were washed inside an Ultrafree centrifugal filter unit (Durapore PVDF 0.65 µm, Millipore) with binding buffer (500 mM NaCl, 50 mM Tris pH 7.8, 5 mM imidazole, 5% (v/v) glycerol). The remaining 45 µL of the RRL expression was added to the beads and incubated for 1 hr at 4°C. Subsequently the beads were washed 5 times with ice-cold wash buffer (500 mM NaCl, 50 mM Tris pH 7.8, 30 mM imidazole, 5% (v/v) glycerol) and the proteins were eluted with 40 µL of 80°C hot SDS-PAGE buffer (P). After SDS-PAGE and western blotting the quotient of pull-down (P) and input (I) band intensity was normalized to TAp63α incubated after expression with His$_6$-tagged p73 TD (set to 1).

## Real-time quantitative PCR

Real-time quantitative PCR was performed with two independent sets of samples. For each condition per set four dissected ovaries were pooled. Oocytes were isolated by trypsin-digestion and multiple centrifugation steps. Total RNA was extracted applying the PicoPure RNA Isolation Kit (Applied Biosystems) with on-column DNAseI (Qiagen) digestion and subsequently subjected to reverse transcription with random primers using the RETROscript Kit (Ambion) followed by cDNA amplification with the TaqMan PreAmp Kit (ThermoFisher Scientific).

Real-time quantitative PCR to determine the fold-induction of p63 target genes was performed with the TaqMan Gene Expression System (ThermoFisher Scientific) using a LightCycler 480 (Roche). For one biological set, each sample and TaqMan assay probe combination was measured in duplicates.

All Kits were used according to the manufacturer's instruction. The following TaqMan assays (ThermoFisher Scientific) were purchased for the preamplification step and the gene expression analysis: TBP (Mm00446971_m1), Msy (Mm01250826_g), p21 (Mm04205640_g1), PUMA (Mm00519268_m1) and Mdm2 (Mm01233136_m1).

Target gene signals were referenced to the house keeping gene TBP and mean fold-induction upon irradiation was calculated for the biological duplicates including error propagation. The significance levels were determined by the student's t-test.

Permission for the experiments with mouse ovaries was obtained from the "Tierschutzbeauftragte" of the Goethe University.

## Size exclusion chromatography (SEC)

Analytical SEC was performed in phosphate buffer (50 mM sodium phosphate pH 7.8, 100 mM NaCl) at 4°C using a Superose 3.2/300 column (GE Healthcare) (injection volume 50 µL; flow rate 50 µL/min; fraction size 50 µL). SEC fractions were quantified by western blotting. Analytical SEC of TAp63α$_{min}$ in urea was performed as described detailed in Supplemental Experimental Procedures.

## Analytical SEC of TAp63α$_{min}$ in presence of urea

SEC experiments were performed on an ÄKTApurifier system at 4°C using a Superpose 6 3.2/300 column (GE Healthcare), monitoring absorption at 280 nm.

## Analytical SEC of TAp63α$_{min}$ at different urea concentrations

The column was equilibrated in a phosphate buffer containing urea at a variable concentration X. 5 µL of TAp63α$_{min}$ (102 mg/mL) were diluted with 75 µL of buffer X (to a final concentration of 6.4 mg/mL) and incubated for one hour at 4°C before being injected on the column. This experiment was performed at different urea concentrations X [M]: 1, 1.25, 1.5, 1.75, 2, 2.25, 2.5, 3, 3.5, 4, 4.5, 5, 6, and 7.

## Analytical SEC of TAp63α$_{min}$ at constant urea concentration

The column was equilibrated in a phosphate buffer containing 1.75 M urea. TAp63α$_{min}$ was diluted to a final concentration of 6.4 mg/mL in a buffer with a final urea concentration of 1.75 M (first TAp63α$_{min}$ was diluted with x µL of buffer X and then with additional y µL of buffer Y, whereby $c_y = c_x + 1$ M, so that the final concentration was exactly 1.75 M). Injections were performed at different time points [hours:minutes]: 0:01, 0:53, 1:45, 2:43, 3:36, 4:30, 5:29, 6:20, 7:17, 8:14, 9:06, 24 hr.

## Analytical SEC followed by dialysis and reinjection

The column was equilibrated in a phosphate buffer containing 1.75 M urea. TAp63$\alpha_{min}$ (32 mg/mL final concentration) was incubated in phosphate buffer with 1.75 M urea for one hour and then injected on the column. The dimer and tetramer peak (two fractions each) was dialyzed back to 0 M urea using D-tube Dialyzer Mini (MWCO 12–14 kDa) in a 50 mL falcon filled with phosphate buffer under continuous stirring. After 13 hr of dialysis the samples were reinjected on the column equilibrated with phosphate buffer.

## Biomolecular structures

We used the crystal structure of the p63 tetramerization domain (PDB: 4A9Z) (*Muniz et al., 2011*) to highlight interactions relevant in context of dimeric TAp63$\alpha$. The crystal structure of the p63 DNA binding domain (DBD) in complex with DNA (PDB: 3QYN) (*Chen et al., 2011*; *Chen and Herzberg, 2011*) was used to model the interaction with ASPP2 by structural alignment with the p53-ASPP2 complex (PDB: 1YCS) (*Gorina and Pavletich, 1996*; *1997*).

All structures and models were illustrated using PyMOL 1.7.6.6.

## TAD/TID dissociation assay

To obtain a qualitative measure of TAD and TID dissociation, constructs with a TEV$_{site}$ (ENLYFQGS) between residues 66 and 67 (591 and 592) and with a C-terminal (N-terminal) myc-tag were created. After RRL expression cycloheximide (50 µg/mL final) and TEV protease (10 µg) were added. The sample was incubated for either 15 min, 1 hr, 4 hr or 12 hr at 37°C before being cooled to 4°C and subsequently analyzed by SEC.

## Transactivation assays

Transcriptional activities of TAp63$\alpha$ and TAp63$\alpha_{(10-614)}$ mutants were measured in triplicates as described previously (*Luh et al., 2013*).

## Western blotting

Western blot (WB) analysis was performed as described previously (*Straub et al., 2010*).

## Small-angle X-ray scattering

In-line size exclusion chromatography small-angle X-ray scattering of TAp63$\alpha_{min}$ was performed at bending magnet beamline B21 at Diamond Light Source (Harwell, UK). The output from an Agilent HPLC was connected to an in-vacuum quartz flow cell. The SAXS detector was triggered by the 280 nm UV sensor in the Agilent HPLC, and allowed the collection of data in 1 s time bins across the peak of interest. A Shodex KW404 column was utilised for these experiments. At the end of each experimental run, SAXS data were integrated using beamline software and the background subtracted using running buffer. The integration procedure ensured that only SAXS data from the peak of interest were abstracted and subjected to further analysis. Data were inspected for radiation damage and aggregation by inspection of Guinier plots. This method ensured that SAXS data were unperturbed by any other oligomers which may have formed or been present in the analysis solution.

The beamline was also used to collect data in batch mode, whereby protein and corresponding buffer solutions were exposed to the beam using an Arinax (Grenoble, France) BioSAXS automated sample changer robot, consisting of temperature controlled storage and exposure units. The exposure unit contained a 1.6 mm diameter quartz capillary in which the samples were illuminated with the x-ray beam; the exposure unit temperature was set to 15°C. The sample capillary was held in vacuum and subjected to a cleaning cycle between each measurement. Samples were stored in 96 well plates at 5°C. A Pilatus 2M two-dimensional detector was used to collect 10 frame exposures of 10 s from each sample and the corresponding buffer. The detector was placed at 3.9 m from the sample, giving a useful q-range of 0.008 Å$^{-1}$ < 0.4 Å$^{-1}$, where q = 4$\pi$ sin $\theta$ / $\lambda$, 2$\theta$ is the scattering angle and $\lambda$ is the wavelength, which was set to 1 Å. Two dimensional data reduction consisted of normalization for beam current and sample transmission, radial sector integration, background buffer subtraction and averaging. Each frame was inspected for the presence of radiation induced protein damage; if this was found to be the case, the frames were not reduced and processed.

Further data analysis, such as scaling, merging and Guinier analysis were performed in Scatter (*Forster et al., 2010*). Three concentrations were measured of each mutant with each experimental data frame being inspected for signs of radiation damage. Frames which appeared to demonstrate radiation damage were excluded from averaging.

Ab-initio shape reconstruction of the wild type was performed by averaging and filtering 13 runs of DAMMIF (*Franke and Svergun, 2009*), with a final refinement in DAMMIN (*Svergun, 1999*), utilizing slow mode. The wild type was found to have $R_g$ of 38.6 Å, with $D_{max}$ of 132 Å. $\lambda$-cro-TAp63$\alpha_{min}$ was analyzed using MONSA, allowing a simultaneous bead modelling from the wild type and the N-terminal fusion. A relative volume difference for MONSA was derived from Porod analysis of the wild type and derivative scattering curves.

## Secondary structure prediction

Secondary structure and disorder were predicted with Phyre2 (*Kelley et al., 2015*) and the Protein Crystal Structure Propensity Prediction Server (*Price II et al., 2009*) which uses PredictProtein (*Rost et al., 2004*).

## Acknowledgements

ASPP2 was a gift from Xin Lu, Oxford. The research was funded by the DFG (DO 545/8-1), the Centre for Biomolecular Magnetic Resonance (BMRZ), and the Cluster of Excellence Frankfurt (Macromolecular Complexes). DC was supported by a Boehringer Ingelheim Fonds PhD Fellowship. The Structural Genomics Consortium is a registered charity (number 1097737) that receives funds from the Canadian Institutes for Health Research, the Canadian Foundation for Innovation, Genome Canada through the Ontario Genomics Institute, GlaxoSmithKline, Karolinska Institutet, the Knut and Alice Wallenberg Foundation, the Ontario Innovation Trust, the Ontario Ministry for Research and Innovation, Merck & Co., Inc., the Novartis Research Foundation, the Swedish Agency for Innovation Systems, the Swedish Foundation for Strategic Research, and the Wellcome Trust J Hannewald is an employee of Merck KGaA, Darmstadt.

## Additional information

### Competing interests

VD: Reviewing editor, *eLife.* The other authors declare that no competing interests exist.

### Funding

| Funder | Author |
| --- | --- |
| Deutsche Forschungsgemeinschaft | Daniel Coutandin<br>Christian Osterburg<br>Ratnesh Kumar Srivastav<br>Manuela Sumyk<br>Sebastian Kehrloesser<br>Jakob Gebel<br>Marcel Tuppi<br>Birgit Schäfer<br>Uta Müller-Kuller<br>Manuel Grez<br>Volker Dötsch |
| Wellcome Trust | Eidarus Salah<br>Sebastian Mathea<br>Stefan Knapp |

The funders had no role in study design, data collection and interpretation, or the decision to submit the work for publication.

### Author contributions

DC, CO, Conception and design, Acquisition of data, Analysis and interpretation of data, Drafting or revising the article; RKS, MS, SKe, JG, MT, JH, BS, ES, SM, UM-K, JD, Acquisition of data, Analysis

and interpretation of data; MG, SKn, Conception and design, Analysis and interpretation of data; VD, Conception and design, Analysis and interpretation of data, Drafting or revising the article

## Author ORCIDs

Volker Dötsch, http://orcid.org/0000-0001-5720-212X

## Ethics

Animal experimentation: The work with mice was conducted according to the regulations of the Goethe University and the DFG (according to § 4 TierSchG) and supervised by the Tierschutzbeauftragte of Goethe University.

# Additional files

## Supplementary files

• Supplementary file 1. Mutagenesis screen of residues on the surface of the DBD. Constructs were expressed in rabbit reticulocyte lysate and subjected to size exclusion chromatography on a Superose 6 3.2/300. All constructs formed dimers indicating that none of these residues are involved in essential contacts inside dimeric TAp63α.

## Major datasets

The following previously published datasets were used:

| Author(s) | Year | Dataset title | Dataset URL | Database, license, and accessibility information |
|---|---|---|---|---|
| Gorina S, Pavletich NP | 1997 | P53-53BP2 COMPLEX | http://www.rcsb.org/pdb/explore/explore.do?structureId=1YCS | Publicly available at the RCSB Protein Data Bank (accession no. 1YCS) |
| Muniz JRC, Coutandin D, Salah E, Chaikuad A, Vollmar M, Weigelt J, Arrowsmith CH, Edwards AM, Bountra C, Dotsch V, Von Delft F, Knapp S | 2011 | CRYSTAL STRUCTURE OF HUMAN P63 TETRAMERIZATION DOMAIN | http://www.rcsb.org/pdb/explore/explore.do?structureId=4A9Z | Publicly available at the RCSB Protein Data Bank (accession no. 4A9Z) |
| Chen C, Herzberg O | 2011 | Structure of p63 DNA Binding Domain in Complex with a 22 Base Pair A/T Rich Response Element Containing 2 Base Pair Spacer Between Half Sites | http://www.rcsb.org/pdb/explore/explore.do?structureId=3QYN | Publicly available at the RCSB Protein Data Bank (accession no. 3QYN) |

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
