## [Decision Letter]

Thank you for submitting your work entitled "Quality control in oocytes by p63 is
based on a spring-loaded activation mechanism on the molecular and cellular level"
for consideration by *eLife*. Your article has been favorably evaluated
by Jessica Tyler (Senior editor) and three reviewers, one of whom is a member of our
Board of Reviewing Editors.

The reviewers have discussed the reviews with one another and the Reviewing Editor has
drafted this decision to help you prepare a revised submission.

In the present manuscript, the authors report, using a nice experimental model, that
activation of the TAp63 transcription factor leads to tetramerization that follows a
spring-loaded mechanism in the absence of any translation of other cellular factors in
oocytes and is associated with unfolding of the inhibitory structure that blocks the
tetramerization interface. This is highly important, as the mechanism of activation of
its more famous family member p53 is totally different. The consequences of this
mechanism are extremely important to elucidate how quality control of oocytes is
achieved. The data presented are of excellent quality and expand the area of TAp63 in
the quality control of oocytes. Indeed, the detailed analysis of TAp63 structural
analysis at the molecular level represents relevant new insight on reproductive biology
as well as in understanding the mechanism of p63 regulation, as compared for example to
p53. Consequently, this is a very important, highly interesting and solid study
investigating the consequences of loss of p63, or its genetic mutations at the
structural level.

However, the reviewers have a number of concerns about some of the data and its
interpretation. Overall, the reviewers agreed to encourage resubmission of a revised
manuscript addressing these concerns.

1) In Figure 6 the argument that p63 activation
is post translationally regulated by phosphorylation is not substantiated by the data.
The authors present an attractive model for activation of TAp63α upon DNA damage in
oocytes. This is supported by demonstration of TAp63 tetramer formation and induction of
p63 target gene expression (p21, Puma, Mdm2) in irradiated murine oocytes. However, this
model is incomplete without data showing that TAp63 phosphorylation indeed triggers
tetramerization and activation. The reviewers propose a number of experiments to address
this issue. First, treatment of sample lysates with a phosphatase to show that the
migration differences post-irradiation is indeed due to phosphorylation. It is possible
that the differences in mobility are due to some other post-translational modification.
Second, if, as they propose in their discussion, ATM or CHECK2 are putative culprits in
activating TAp63, this needs to be demonstrated in some fashion, either via genetic or
pharmacological inactivation of these kinases. This could also be tested in vitro by
adding relevant kinases, i.e. ATM and/or CHEK2. In addition, specific phosphorylation
sites in TAp63α could be mutated and the effect on tetramerization could be examined in
vitro and in oocytes. This type of data would strengthen the manuscript
considerably.

2) In the subsection “Defining the minimal sequence required for formation of the closed
dimeric conformation”, the authors refer to three different constructs:
TAp63α_min_, wild type TAp63α, and a slightly shortened version
TAp63α_(10-614)_. Two constructs are shown in Figure 1 and Figure 1—figure supplement 2. It looks as if the slightly shortened version,
TAp63α_(10-614)_, is not shown. Yet the TAp63α_min_ construct
contains exactly residues 10-614. This should be clarified. Which construct is
shown?

3) In Figure 6, levels of TAp63 tetramers in
irradiated oocytes are significantly lower than those of dimeric TAp63 prior to
irradiation, probably less than 10%. This is apparent both on the native gel (B) and the
denaturing SDS gel (C). Thus, it seems like the formation of tetramers is associated
with a dramatic reduction in TAp63 protein levels. This is not discussed. What is the
reason for the much lower levels of TAp63 tetramers? p63 target genes are induced in
irradiated oocytes (Figure 6). Are p53 and/or p73
expressed in these cells upon DNA damage? Did the authors exclude the possibility that
the observed induction of target genes is due to activation of p53 or p73?

4) It would be important to measure caspase activation in response to ectopic p63
expression in SAOS cells using the p63 mutants.

---

## [Author Response]

*1) In Figure 6 the argument that p63
activation is post translationally regulated by phosphorylation is not substantiated
by the data. The authors present an attractive model for activation of
TAp63α*

*upon DNA damage in oocytes. This is supported by demonstration of TAp63 tetramer
formation and induction of p63 target gene expression (p21, Puma, Mdm2) in irradiated
murine oocytes. However, this model is incomplete without data showing that TAp63
phosphorylation indeed triggers tetramerization and activation. The reviewers propose
a number of experiments to address this issue. First, treatment of sample lysates
with a phosphatase to show that the migration differences post-irradiation is indeed
due to phosphorylation. It is possible that the differences in mobility are due to
some other post-translational modification. Second, if, as they propose in their
discussion, ATM or CHECK2 are putative culprits in activating TAp63, this needs to be
demonstrated in some fashion, either via genetic or pharmacological inactivation of
these kinases. This could also be tested in vitro by adding relevant kinases, i.e.
ATM and/or CHEK2. In addition, specific phosphorylation sites in TAp63α could be
mutated and the effect on tetramerization could be examined in vitro and in oocytes.
This type of data would strengthen the manuscript considerably.*

The reviewers wanted to see a direct link between phosphorylation and activation and
asked to provide evidence that the shift seen on the SDS gel of TAp63α following
irradiation is due to phosphorylation. We had provided such evidence already in the Cell
paper (Deutsch et al., 2011). There we have treated oocyte lysates after irradiation
showing both tetramerization as well as a shift on an SDS gel with λ-phosphatase.
Despite the complete removal of the shift on the SDS gel the gel filtration profile of
the now non-phosphorylated TAp63α remains virtually the same, indicating that 1) the
shift is indeed due to phosphorylation and 2) that phosphorylation is only the trigger
to induce the change from a dimeric to a tetrameric state, supporting our analysis that
the dimeric state is a high energy kinetically trapped one. To make this more clear we
have added the citation of our Cell paper to the section in which we have mentioned
these experiments in the original version of the manuscript (“The dimeric conformation
of TAp63α constitutes a kinetically trapped state”) and have added a few sentences in
the Discussion part.

In addition, the reviewers asked us to prove that phosphorylation triggers activation
and tetramerization. Activation through phosphorylation by Chk2 has been shown in the
paper by Bolcun-Filas et al. (Science 2014) that we cite several times. They had,
however, not linked activation to tetramerization. To make this link we have added
experimental results to Figure 6 that show that
addition of 25 µM of a Chk2 inhibitor results in the suppression of phosphorylation and
tetramerization. Furthermore, these experiments demonstrate that inhibiting the
activation of TAp63α also results in stabilization of the protein. This is consistent
with earlier investigations that had shown a link between proteasomal degradation and
active DNA binding and transactivation. The observation that inhibition of activation
results in higher cellular protein levels of TAp63α also supports our model that in the
inhibited state the TAD is not accessible and only the open state can be degraded fast.
We discuss all of this in a new paragraph at the end of the Results section.

*2) In the subsection “Defining the minimal sequence required for formation of
the closed dimeric conformation”, the authors refer to three different constructs:
TAp63α_min_, wild type TAp63α, and a slightly shortened version
TAp63α_(10-614)_. Two constructs are shown in Figure 1 and Figure 1—figure supplement 2. It looks as if the slightly shortened version,
TAp63α_(10-614)_, is not shown. Yet the TAp63α_min_ construct
contains exactly residues 10-614. This should be clarified. Which construct is
shown?*

The reviewers asked to clarify the exact boundaries of the different constructs used. We
have modified Figure 1—figure supplement 2 now
showing all three different versions of TAp63α used in the experiments reported in this
manuscript.

*3) In Figure 6, levels of TAp63 tetramers
in irradiated oocytes are significantly lower than those of dimeric TAp63 prior to
irradiation, probably less than 10%. This is apparent both on the native gel (B) and
the denaturing SDS gel (C). Thus, it seems like the formation of tetramers is
associated with a dramatic reduction in TAp63 protein levels. This is not discussed.
What is the reason for the much lower levels of TAp63 tetramers? p63 target genes are
induced in irradiated oocytes (Figure 6). Are
p53 and/or p73 expressed in these cells upon DNA damage? Did the authors exclude the
possibility that the observed induction of target genes is due to activation of p53
or p73?* The reviewers asked about the reduced cellular levels of TAp63α
following activation. As explained above this is due to fast proteasomal degradation of
active p63 (see response to question #1). We have also investigated if p53 or p73 can be
detected before or after irradiation in oocytes. Using immunohistochemistry, we have not
been able to detect p53 and only a weak and diffuse staining for p73. The much higher
concentration of p63 together with the fact that significant induction of target genes
is achieved in the presence of the translation inhibitor cycloheximide (preventing an
increase of the cellular content of p73 through transcription by p63) strongly argues
that the transcriptional effects seen are due to p63. We describe these results in the
last paragraph of the Results section and provide the data in Figure 6—figure supplement 1.

4) It would be important to measure caspase activation in response to ectopic
p63 expression in SAOS cells using the p63 mutants.

We have tried to show caspase activation in response to ectopic expression of activated
p63 mutants in different cell lines (SAOS, H1299, SUM159) but were unable to detect
apoptosis in these cell lines (no cleaved PARP or activated caspase-3). This might be
due to effective mechanisms of these cancer cell lines to suppress apoptosis or that
additionally DNA damage is necessary to activate the apoptotic program.

Author response image 1.Caspase assay.H1299 (**A**), SUM159 (**B**) and SAOS2 (**C**)
cells were transfected with TAp63α mutants. Caspase-3 Antibody (Cell Signaling
#9662) and PARP Antibody (Cell Signaling #9542) were used in A and B to detect
full-length proteins and cleaved fragments (large fragment of caspase-3 (17
kDa) and large fragment of PARP (89 kDa)). Cleaved Caspase-3 Antibody (Cell
Signaling #9662) was used in C to detect cleaved Caspase-3. No signal of any
cleaved fragment was detected in dependency of TAp63α mutant transfection. Only
inactive TAp63α (either wildtype or mutants bearing mutation F16A W20A L23A or
R304Q (DNA-binding deficient)) reached high expression levels in the cells.**DOI:**
http://dx.doi.org/10.7554/eLife.13909.025